# Screening the Marker Components in *Psoralea corylifolia* L. with the Aids of Spectrum-Effect Relationship and Component Knock-Out by UPLC-MS^2^

**DOI:** 10.3390/ijms19113439

**Published:** 2018-11-02

**Authors:** Mengjun Shi, Yan Zhang, Miaomiao Song, Yong Sun, Changqin Li, Wenyi Kang

**Affiliations:** 1National Center for Research and Development of Edible Fungus Processing Technology, Henan University, Kaifeng 475004, China; 18237879687@163.com (M.S.); snowwinglv@126.com (Y.Z.); smm2985930@163.com (M.S.); sy183509@163.com (Y.S.); 2Kaifeng Key Laboratory of Functional Components in Health Food, Kaifeng 475004, China; 3Hebei Food Inspection and Research Institute, Shijiazhuang 050091, China; 4Beijing Academy of Food Sciences, China Meat Research Center, Beijing 100068, China

**Keywords:** *Psoralea corylifolia*, spectrum-effect relationships, component knock-out, tyrosinase

## Abstract

*Psoralea corylifolia* L., (*P. corylifolia*), which is used for treating vitiligo in clinic, shows inhibitory and activating effects on tyrosinase, a rate-limiting enzyme of melanogenesis. This study aimed to determine the active ingredients in the ethenal extracts of *P. corylifolia* on tyrosinase activity. The spectrum-effect relationship and knock-out method were established to predict the active compounds. Their structures were then identified with the high resolution mass spectra. A high performance liquid chromatography method was established to obtain the specific chromatograms. Tyrosinase activity in vitro was assayed by the method of oxidation rate of levodopa. Partial least squares method was used to test the spectrum-effect relationships. Chromatographic peaks P2, P4, P9, P10, P11, P13, P21, P26, P28, and P30 were positively related to the activating effects on tyrosinase activity in PE, whereas chromatographic peaks P1, P3, P6, P14, P16, P19, P22, and P29 were negatively related to the activating effects on tyrosinase in the *P. corylifolia* (PEs). When the sample concentration was 0.5 g·mL^−1^, equal to the amount of raw medicinal herbs, the target components were daidzein (P2), psoralen (P5), neobavaisoflavone (P13), and psoralidin (P20), which were consistent with the results of spectrum-effect relationships.

## 1. Introduction

Vitiligo, a clinically common disease, is caused by acquired, localized, or generalized skin depigmentation. The local skin of patients suffers from certain degree of melanocytic function loss [1]. Tyrosinase (TYR) is a rate-limiting enzyme that is involved in the biosynthesis of melanocytes and thus, activation of TYR can increase the production of melanin and can be applied for treatment of vitiligo [2]. As described in 2015 edition of the Chinese Pharmacopoeia, *Psoralea corylifolia* L. (*P. corylifolia*) was externally used to treat vitiligo and alopecia areat [3].

Dry and mature fruit of *P. corylifolia*, which belongs to leguminous plant family and is used as Traditional Chinese Medicine (TCM) in clinic, is one of the most commonly used TCMs for the treatment of vitiligo. According to their chemical structures, about 90 compounds, including coumarins, flavonoids, meroterpenes, and benzofuran, have been isolated and identified [4]. Furocoumarin compounds in *P. corylifolia*, either coated or taken orally, coupled with the sun exposure can result from the aggregation of photosensitization pigment. One of the underlying mechanisms is that the skin absorbs light energy and combines with DNA to form an optical adduct, increasing TYR activity and melanin synthesis [5]. Other studies have shown that psoralen and bavachinin A can inhibit melanin synthesis by inhibiting both TYR activity and enzyme content in A375 cells [6,7]. Li et al. also found that bakuchiol at the low concentration had stronger inhibiting effect on TYR than that of arbutin [8].

Therefore, there is a bi-directional regulation of TYR activity in *P. corylifolia*, which is closely related to its extraction parts, heating treatment, and concentration [9,10,11]. There may be a component in *P. corylifolia* that inhibits and activates TYR activity. In order to illustrate the effect of components in *P. corylifolia* on TYR activity, the spectrum-efficiency model was established by using the analysis method of high performance liquid chromatography (HPLC) to establish fingerprint and a TYR model in vitro to conduct an appropriate efficacy evaluation, coupled with data processing methods [12].

Based on the active compounds screened out from the spectrum-efficiency model, the method of component knock-out [13] and chromatographic knockout technique were established [14,15] to prepare the target components and their corresponding negative samples, which were evaluated based on the effects of the target components, negative samples, and total extracts on TYR activity. In this study, we aimed to determine the accuracy of spectrum-effect relationship and to reveal the interaction between the active components of TYR in *P. corylifolia* by spectrum-efficiency model, which was used to determine the key components in *P. corylifolia* for the treatment of vitiligo, and, at the same time, to explore the interactions between the various components.

Recent researches have shown that a large number of drugs act on enzymes, which are important drug targets. Furthermore, many amazing technologies and ideas have been available and applied to screen out the related active drugs. Among them, the most commonly used methodologies are high throughput screening technology [16] and structure-activity relationship analysis technology [17,18]. Our present study used the ideas of the previous technology and adjusted them. By re-positioning the spectrum-effect relationships and verifying them with the component knock-out method, we realized the isolation of the different components from *P. corylifoli* and examined their effects on TYR.

## 2. Results and Discussion

### 2.1. Spectrum-Effect Relationships

#### 2.1.1. Quantitative Determination of Chromatographic Peaks

Fingerprint spectra are a recognized method used worldwide to evaluate the quality of traditional Chinese medicines (TCMs). In this study, we used fingerprint spectra to circumvent the weaknesses of the fingerprint technique and foster the strengths in TCM and to investigate spectrum-effect relationships of TCMs [19]. The study of spectrum-effect relationship can reveal the material basis of the therapeutic values of TCMs. This method has been widely applied in the researches of pharmacodynamic components of Chinese medicine [20], the compatibility of components [21], processing mechanism and the prediction of efficacy [22], the optimization of process [23], and the screening of toxic components of TCMs. The spectrum-effect relationship researches were conducted based on different fields, origins, harvest times, processing methods and batches of TCMs [24,25,26,27,28,29]. Therefore, ten different batches of samples were selected to establish the fingerprint in order to accurately determine the material basis of their effects on tyrosinase activity.

“Chinese traditional medicine chromatographic fingerprint similarity evaluation system (2004, 1.0 A Edition)” was used to quantitatively determine the chromatographic peaks. Multi-point correction of chromatographic peak position was performed based on chromatographic peaks, which were found in each sample with good separation by reference to the chromatogram of PEs. A contrast chromatogram was generated by the average method. The matching chromatograms and peak areas were shown in Figure 1 and Appendix A.

#### 2.1.2. Determination of Activation Effect of PEs on Tyrosinase Activity In Vitro

The activation rates of PEs from different batches of *P. corylifolia* on tyrosinase activity were shown in Table 1. When the concentration of the sample was changed, the effect on the tyrosinase activity showed a bi-directional pattern.

When the concentration of S5 was 0.0625 g·mL^−1^, equal to the amount of raw medicinal herbs, it showed inhibitory effect on tyrosinase activity. When the concentration of S8 was 1 g·mL^−1^, equal to the amount of raw medicinal herbs, it showed inhibitory effect on tyrosinase activity. When the concentrations of S5 were 0.2500, 0.1250, and 0.0625 g·mL^−1^, equal to the amount of raw medicinal herbs, it showed an inhibitory effect on tyrosinase activity. When the sample concentration was 1 g·mL^−1^, equal to the amount of raw medicinal herbs, all the other nine batches but S8 of samples caused activating effects on tyrosinase activity. The activation effects of sample S1 on tyrosinase activity was the strongest. When compared to S1, extremely significant differences were seen among the other nine batches of samples (*p* ≤ 0.001). It indicated that the activation degrees of 10 batches of samples on tyrosinase were quite different at this concentration. When the sample concentration was 0.5 g·mL^−1^, equal to the amount of raw medicinal herbs, the 10 batches of samples activated tyrosinase activity. The activation of tyrosinase was the strongest in sample S4 at this concentration. When compared with S4, extremely significant differences were seen the other nine batches of samples (*p* ≤ 0.001). It indicated that the activation degree of 10 batches of samples on tyrosinase was quite different at this concentration. When the sample concentration was 0.25 g·mL^−1^, equal to the amount of raw medicinal herbs, the other nine batches of samples, except S9 were active on tyrosinase. The activation of tyrosinase was the strongest in sample S5 at this concentration. Compared with S5, the other nine batches of samples exhibited extremely significant differences (*p* ≤ 0.001). It indicated that the activation degree of 10 batches of samples on tyrosinase was quite different at this concentration. When the sample concentration was 0.125 g·mL^−1^, equal to the amount of raw medicinal herbs, the other nine batches of samples except S9 were active on tyrosinase. The activation of tyrosinase was the strongest in sample S8 at this concentration. The activation rates of S6, S10, and S8 on tyrosinase were equivalent (*p* > 0.5), while the other batches were with significant difference (*p* ≤ 0.001) as compared with S8. When the sample concentration was 0.0625 g·mL^−1^, equal to the amount of raw medicinal herbs, the other eight batches of samples were active on tyrosinase in addition to S5 and S9. Among them, the activation of tyrosinase was the strongest in sample S8 at this concentration. The activation rates of S6 and S7 on tyrosinase were equivalent (*p* > 0.5), while those of other batches were significantly different, as compared with S6 (*p* ≤ 0.001).

Therefore, with the change in the concentration of samples, the different batches of samples exhibited different trends on tyrosinase activity and the bidirectional regulation, due to the composition of *P. corylifolia* (PE), which caused different effects on tyrosinase.

#### 2.1.3. Regression Equation of Partial Least Squares Analysis

After many years of development and exploration, the research pattern of spectrum-effect relationships has basically formed. The establishment of fingerprint, pharmacodynamic evaluation, and data processing of spectrum-effect relationship are the important components of the research [30]. The most commonly used methods for the establishment of fingerprints are HPLC, UPLC, GC, and GC-MS [31]. Pharmacodynamics often used in vitro or in vivo experimental models to obtain “effective” information [32]. The most commonly used data processing methods include principal component analysis (PCA), canonical correlation analysis (CCA), partial least squares analysis (PLSR), grey relational analysis (GRDA), and others [33]. The PLSR method is practical and stable, and can contain all the original fingerprint peaks, of which, the information about the reaction is more comprehensive [34].

In our study, the areas of quantitative chromatographic peaks were set as the independent variable (*X*), activation rate of tyrosinase activity (equivalent to the raw material 0.5 g·mL^−1^) was taken as the dependent variable (*Y*). Then, DPS 7.05 statistical software was used for the mean of the data processing and the partial least squares regression analysis. The result showed that when the potential factor reached 7, the R^2^ reached the maximum, and the variance of the explanatory variable was 91.82%. The regression Equation (1) was expressed, as follows:(1)$$\begin{array}{l} Y1\;=\;0.000036\;-\;0.350240X1\;+\;0.217858X2\;-\;0.225877X3\;+\;0.259945X4\;-\;0.015104X5\;-\;0.197908X6\;+\;0.044606X7\;+\;0.058149X8\;+\;0.252581X9\;\\ \;+\;0.337107X10\;+\;0.149909X11\;-\;0.042169X12\;+\;0.288226X13\;-\;0.307045X14\;-\;0.085881X15\;-\;0.201229X16\;+\;0.145773X17\\ \;-\;0.056096X18\;-\;0.249421X19\;-\;0.058010X20\;+\;0.209306X21\;-\;0.351590X22\;-\;0.106859X23\;-\;0.085872X24\;-\;0.010733X25\\ \;+\;0.267864X26\;-\;0.073532X27\;+\;0.195020X28\;-\;0.199972X29\;+\;0.362664X30 \end{array}$$

The regression coefficients of the partial least squares regression equation was shown in Figure 2. Chromatographic peaks P2, P4, P9, P10, P11, P13, P21, P26, P28, and P30 were positively related to their activation effects on tyrosinase in PE and their correlation coefficients were larger (|*R*| > 0.2), meaning that when the content of the compounds, of which, these peaks represented were increased, the activation on tyrosinase would be stronger. Chromatographic peaks P1, P3, P6, P14, P16, P19, P22, and P29 were negatively related to the activation effects on tyrosinase in PE and the correlation coefficient was larger (|*R*| > 0.2), meaning that when the content of the compounds, of which these peaks represented, were increased, the activation on tyrosinase would be weaker.

### 2.2. Component Knock-Out Methods

#### 2.2.1. HPLC Chromatogram of Knock-Out Components and Negative Samples

The target components and negative samples were prepared by high performance liquid chromatography (HPLC), as shown in Figure 3 (the rest is in the Appendix A). According to the peak area normalization method, the purity of each target component was higher than 85%, and the separation degree was good (*R* > 1.5). The negative sample did not contain the target component.

#### 2.2.2. Component Identification

The negative ion mode is more appropriate for identification of compounds in *P. corylifolia* than is the positive ion mode*.* The HPLC four stage rod-electrostatic field orbit trap higher solution mass spectrometry analysis results of knocked-out components were shown in Figure 4. Protonated ions, adducted ions, and fragment ions of each peak were then elucidated in accordance with reference HPLC chromatogram of standard solutions and related data in the literatures [35,36,37,38,39,40,41]. The formula of compound P2 with [M + H]^+^ ions at *m*/*z* 254, C_15_H_10_O_4_, produced ion fragments of 136/118, and were tentatively characterized as daidzein (Figure 5a). Psoralen (P5) produced ions at *m*/*z* 186, 159, 143, 131, and 115 by loss of CO, CO_2_, 2CO, and (CO_2_ + CO), respectively, representing the fragmentation pattern of coumarin with an isopentene group (Figure 5b). Neobavaisoflavone (P13) gave product ions at *m*/*z* 267 [M + H − 56]^+^ and 255 [M + H − 68]^+^, corresponding to the losses of C_4_H_8_ and C_5_H_8_, which may exemplify the fragmentation characteristics of prenyl isoflavones (Figure 5c). The formula of compound P18 with [M + H]^+^ ions at *m*/*z* 320, C_20_H_16_O_4_, produced ion fragments of 279 and 137, corresponding to the losses of C_4_H_8_ and C_7_H_4_O_3_, respectively, and were tentatively characterized as corylin (Figure 5d). In the MS/MS analysis (Figure 5e), product ions at *m*/*z* 336, 281 and 253 by loss of C_4_H_8_ and CO were found to be the typical fragments of psoralidin (P20). Isobavachalcone (P23) with [M + H]^+^ ions at *m*/*z* 324, C_20_H_16_O_5_, is a representative compound of chalcones, featuring the loss of C8H8O, which may be triggered by the retro–Diels–Alder reaction (Figure 5f). Bavachinin A (P24) with [M + H]^+^ ions at *m*/*z* 338, C_22_H_26_O_4_, produced ion fragments of 283 and 219, corresponding to the losses of C_4_H_8_ (Figure 5g). Accordingly, all of these rules may shed light on the identification of compounds with similar structures in future studies. In conclusion, seven knock-out compounds were identified as daidzein, psoralen, neobavaisoflavone, corylin, psoralidin, isobavachalcone, and bavachinin A, as is consistent with the retention time of standards in HPLC.

### 2.3. Effect of Knocked-Out Components and Negative Samples on Tyrosinase Activity In Vitro

As shown in Table 2, when the sample concentration was 0.5 g·mL^−1^, equal to the amount of raw medicinal herbs, P2, P6, P13, P14, P18, P23, P24, P25, P27, and P28 displayed the activation effects on tyrosinase, of which, P23 had stronger activation effect on tyrosinase. The target components of P2, P13, and P28 are consistent with the results of “spectrum-effect relationships” analysis. Chromatographic peaks P5, P20, and P30 had inhibitory effects on tyrosinase. The target components of P5 and P20 were consistent with the results of “spectrum-effect relationships” analysis.

As shown in Figure 6a, the target components of P2, P6, P23, and P24 and their corresponding negative samples all played an activation effects on tyrosinase, which was consistent with the action direction of total ethanol extract. The tyrosinase activation rates of components and corresponding negative samples were more than those of the total ethanol extract, meaning that there may be antagonism effect between the knock-out components and their corresponding negative samples on tyrosinase activation.

As shown in Figure 6b, the target components of P20 and P30 and their corresponding negative samples had inhibitory effect on tyrosinase, which were contrary to the action direction of the total ethanol extract. It indicated that the target components were strongly antagonistic to the inhibition of tyrosinase components in the negative samples, resulting in the activation of tyrosinase in the total ethanol extract, which was dominant, thus causing the phenomenon that the activation effect of the total ethanol extract on tyrosinase was stronger.

As shown in Figure 6c, the target components of P13, P14, P18, P25, and P28, and their corresponding negative samples had activation effects on tyrosinase, which were consistent with the action direction of total ethanol extract. The sum of activation rates of target components and the negative samples on tyrosinase activation were lower than that of the total ethanol extract, suggesting that there may be a synergistic effect between the knock-out component and corresponding negative samples.

The target component of P5 had inhibitory effect on tyrosinase, but its corresponding negative sample showed activation effect on tyrosinase. Furthermore, the sum of tyrosinase activity of target components and negative samples of the tyrosinase activity was greater than that of the total ethanol extract. It could be inferred that there are components that have inhibitory effects on tyrosinase in negative samples, and that these components are synergistic with the inhibition of tyrosinase.

The biosynthesis of melanin, the primary determinant of human skin color, was catalyzed by tyrosinase. The excessive formation and accumulation of melanin in the skin can cause hyperpigmentation skin disorders, such as melasma, freckles, and geriatric pigment spots [42]. Thus, tyrosinase plays an important role in melanin synthesis and neuromelanin formation. Coumarin and phenolic compounds have been revealed to be the stimulating integumental pigmentation and treatment of vitiligo [43,44]. In this paper, we identified components in *P. corylifolia* that can inhibit or activate tyrosinase activity. It was useful for the pertinence of *P. corylifolia* to treat vitiligo.

## 3. Materials and Methods

### 3.1. Materials

A LC-20AT HPLC system was obtained from Shimadzu (Kyoto, Japan), and was equipped with a degasser, a quaternary gradient low pressure pump, the CTO-20A column oven, a SPD-M20A UV-detector and a SIL-20A automatic sampler. Chromatographic separations of target analytes were performed on a RP-18 endcapped column (4.6 mm × 250 mm, 5 μm). The Microplate Reader was obtained from Multiskan MK3 (Thermo Electron, New York, NY, USA). The Mass spectrometer contained Q-Exactive four stage rod-orbit trap LC-MS/MS system purchased from Thermo Fisher Scientific (Waltham, MA, USA) and Thermo Ultimate 3000 UHPLC system.

Acetonitrile was chromatographic grade. Glacial acetic acid was analytical grade. The pure water was purchased from Hangzhou Wahaha Baili Food Co., Ltd., (Hangzhou, China). (L-3-(3, 4-Dihydroxyphenyl) alanine was obtained from Alfa Aesar (Shanghai, China). Tyrosinase was obtained from Worthington Biochemical Corporation (Lakewood, NJ, USA).

### 3.2. Plant Materials

A total of 10 batches of *P. corylifolia* were collected and listed in Table 3. All of the samples were identified by Professor Changqin Li (Joint International Research Laboratory of Medicine & Food, Henan Province, Henan University).

### 3.3. Extraction

The 10 batches of *P. corylifolia* were crushed by 40 meshes and then put in a test tube after being weighted precisely, and 10 times amount of 70% ethanol (*w*/*v*) was added. Total ethanol extract of *P. corylifolia* (PE) was extracted by maceration for three times, 48 h, 48 h, and 24 h, respectively. Finally, the filtrates were dried, and then methanol solution was added. The concentration was equivalent to the amount of raw medicinal herbs at the concentration of 1 g·mL^−1^ (diluted the concentration in a test).

### 3.4. HPLC Analysis

All of the solutions were filtered through the 0.22 μm microporous membrane before they were injected into the HPLC system. The mobile phase was a mixture of acetonitrile (A)-water (B). The gradient elution steps were set as shown in Table 4. The flow rate was set at 0.8 mL·min^−1^ and the column temperature was 30 °C. The UV detection wavelength was set at 254 nm with the sample volume of 10 μL.

### 3.5. Tyrosinase Inhibition Assay In Vitro

PEs was diluted into a corresponding concentration with methanol, stored at 4 °C in refrigerator, and subsequently used for the determination of the enzyme.

Tyrosinase activation assay was performed in a 96-well microplate format according to the method reported [10]. The compounds were screened for the activation effects on tyrosinase activity using levodopa (LOP) as substrate. The volume of whole reaction system was 100 μL, and performed three holes for each sample. 45 μL of K-phosphate buffer (pH 6.8), and 25 μL of mushroom tyrosinase (TYR, with double distilled water solution concentration of 0.2 U·mL^−1^) were incubated with 5 μL of sample solution at 30 °C for 10 min in water-jacket thermostatic incubator (Sumsung GRP-9270, Shanghai, China). Then, 25 μL of LOP (with double distilled water solution at concentration of 0.5 mmol·L^−1^) was added to the reaction mixture and incubated at 30 °C for 5 min. The enzymatic reaction was monitored by measuring the change in absorbance at 492 nm (A_492_) at 30 °C due to the formation of the dihydroxyphenylalanine (DOPA) chrome for 5 min. The percentage of activation of the enzymatic activity was calculated, as follows: tyrosinase activation activity was expressed as activation rate at a certain concentration. The activation rates (%) were calculated according to the formula as follows:(2)$$\;\mathrm{Activation}\;\mathrm{rate}\;\left(\text{\%}\right)\;=\;\left[\frac{\left(A_{\mathrm{Sample}\;+\;\mathrm{LOP}\;+\;\mathrm{TYR}}{\;-\;A}_{\mathrm{Me}\;+\;\mathrm{LOP}}\right)}{{(A}_{\mathrm{Sample}\;+\;\mathrm{lop}\;+\;\mathrm{TYR}}{\;-\;A}_{\mathrm{Sample}\;+\;\mathrm{LOP})}}-1\right]\times \;100\text{\%}$$

### 3.6. Partial Least Squares Analysis

“Chinese traditional medicine chromatographic fingerprint similarity evaluation system 2004, 1.0 A Edition” was used to correct the retention time of each peak. The peak area was processed by equalization to obtain the quantitative data. The partial least squares regression equation was established with the analysis software DPS 7.05, and the peak area was set as the independent variable (*X*). Tyrosinase activation rate was taken as the dependent variable (*Y*). Chromatographic peaks, which were significantly correlated with activation effects on tyrosinase activity, were determined, respectively.

### 3.7. Knock-Out Method

Under the optimized chromatography conditions that are described in section “HPLC analysis”, PE was prepared at 1 g·mL^−1^ equivalent to raw medicinal herbs. Injection volume was 20 μL every time. The chromatogram at 254 nm was recorded. The eluent solution of the target component and negative solution without the target component were collected according to the peak retention time from the spectrum-effect relationship analysis, respectively. Each component was prepared and eluted in a 15-fold series. Target component solution and negative solution were combined, respectively, dissolved with 0.3 mL of methanol solution, and filtered through 0.22 μm microporous membrane. The filtrated solution contained the target component (denoted as Px+) and the corresponding negative sample (denoted as Px−).

### 3.8. Mass Spectrometry Analysis

The compounds were detected using QExactive four stage rod-orbit trap LC-MS/MS system, including Thermo Ultimate 3000 UHPLC system and QExactive (Thermo Fisher Scientific). Separation was performed with a Waters BEH C18 column (2.1 mm × 100 mm, 1.7 μm; Waters). The mobile phase was a mixture of 0.1% formic acid-water (A) and acetonitrile (B), with an optimized linear gradient elution as follows: 0–5 min: 10% B; 5–30 min: 10–95% B; 30–55 min:95% B; 55–56 min: 95–10% B; and, 56–61 min: 10% B. The flow rate was 0.3 mL·min^−1^. The injection volume was 0.2 μL. The column temperature was set at 25 °C.

Compounds were analyzed with the full scan data in positive ion modes to provide the complementary information for structural identification under the following mass spectrometry conditions: sheath gas flow rate, 35arb; auxiliary gas flow rate, 10arb; spray voltage, 3.5 kV; capillary temperature, 320 °C. Full scan: a scanrange, *m*/*z* 100–1500 and a resolving power, 70,000; The automatic gain control (AGC) was set at 3 × 10^6^.

In addition to the full scan acquisition method, for confirmatory purpose, a targeted MS/MS analysis was also performed using the mass inclusion list and the expected retention times of the target analytes, with solving power of 17,500. The AGC target was set to 10^5^, with the maximum injection time of 50 ms. The isolation window was set at 4.0 *m*/*z*. Collision energy was optimized at 30 eV.

## 4. Conclusions

In this study, the components of *P. corylifolia* that had inhibitory or activating effects on tyrosinase activity were investigated by using the spectrum-effect relationship and component knock-out method. The result showed that the correlations of these components to inhibitory effects on tyrosinase activity were different, and that there were either synergetic or antagonistic effects among these components. When the concentration of each sample was 1 g·mL^−1^ equal to the amount of raw medicinal herbs, psoralen, corylin, psoralidin, isobavachalcone, and bavachinin A showed the inhibitory effects on tyrosinase activity, whereas daidzein and neobavaisoflavone showed the activation effects of tyrosinase activity. The result of daidzein is in agreement with those reported in literatures [42,43], in which the rationality of experimental method and the reliability of the results were verified strongly.

## Figures and Tables

**Figure 1 ijms-19-03439-f001:**
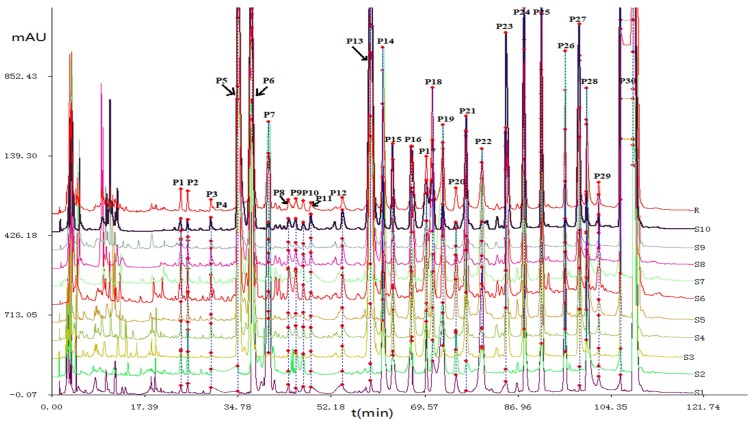
The matching high performance liquid chromatography (HPLC) characteristic chromatograms of different batches of *P. corylifolia.*

**Figure 2 ijms-19-03439-f002:**
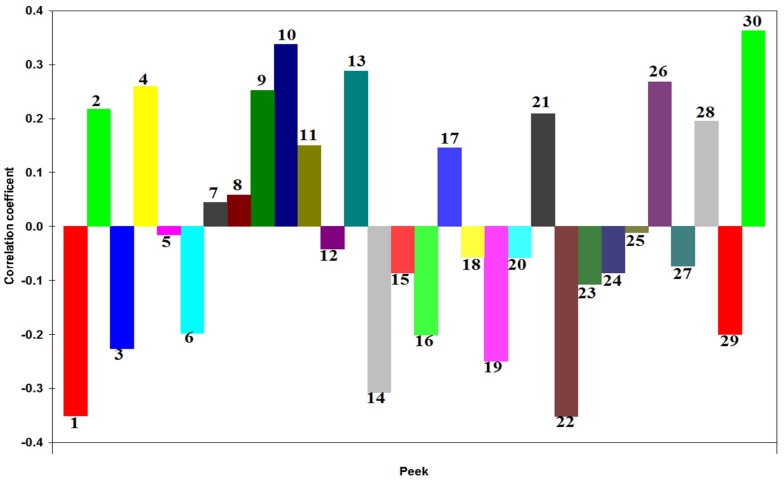
Standardization regression coefficients of partial least squares analysis (PLSR) equations of *P. corylifolia*.

**Figure 3 ijms-19-03439-f003:**
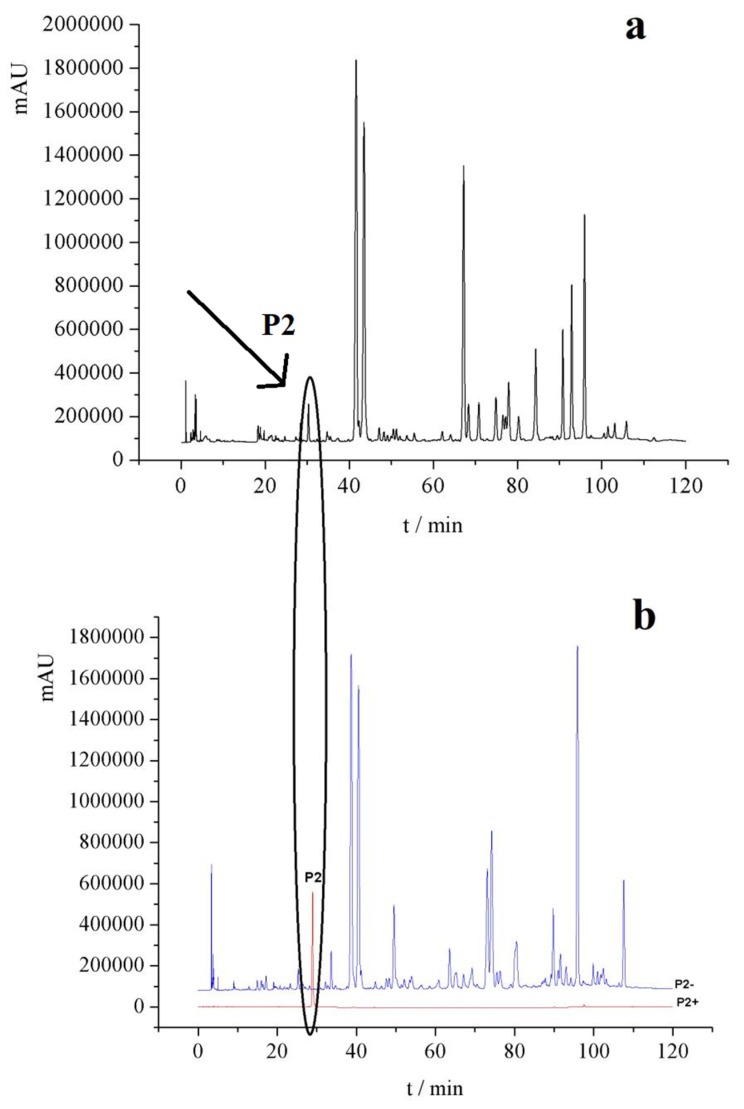
(**a**) HPLC chromatogram of ethanol extract of *P. corylifolia*; (**b**) A component (Px+) and negative samples (Px−).

**Figure 4 ijms-19-03439-f004:**
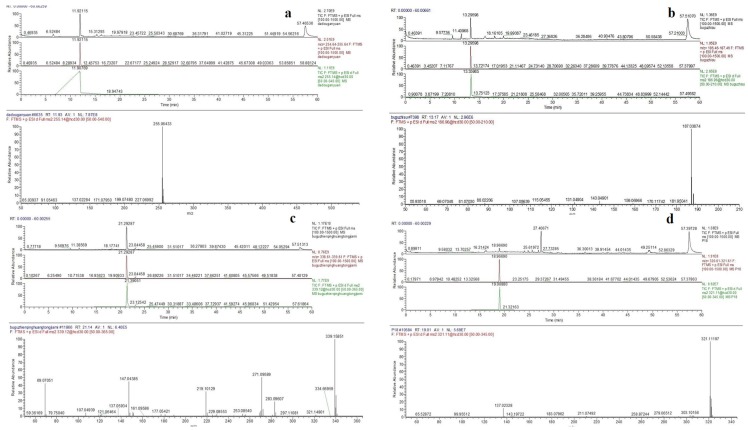
The high resolution mass spectra of (**a**) P2, (**b**) P5, (**c**) P13, (**d**) P18, (**e**) P20, (**f**) P23, and (**g**) P24 knocked-out components.

**Figure 5 ijms-19-03439-f005:**
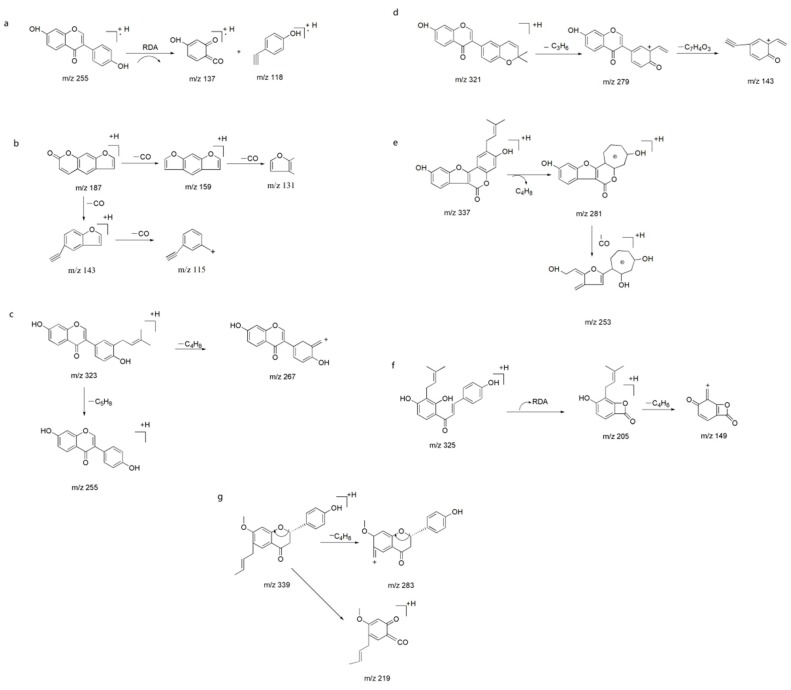
The proposed fragmentation pathways of (**a**) daidzein, (**b**) psoralen, (**c**) neobavaisoflavone, (**d**) corylin, (**e**) psoralidin, (**f**) isobavachalcone, and (**g**) bavachinin A.

**Figure 6 ijms-19-03439-f006:**
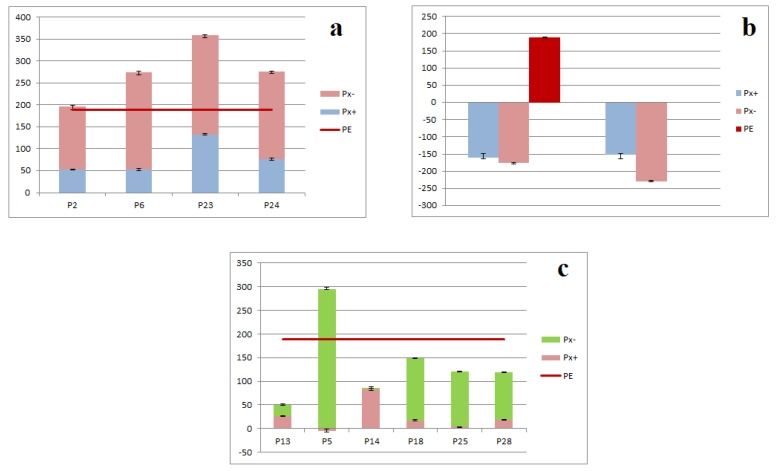
Effect between knocked-out components and negative samples of PE: (**a**) Antagonistic effect on tyrosinase inhibition effect; (**b**) Antagonistic effect on tyrosinase inhibition effect; and, (**c**) Synergetic effect.

**Table 1 ijms-19-03439-t001:** Activation rates of tyrosinase in ethanol extracts of different batches of *P. corylifolia*.

Sample	Concentration of Ethanol Extract (Equal to the Amount of Raw Medicinal Herbs)				
	1 g·mL^−1^	0.5 g·mL^−1^	0.25 g·mL^−1^	0.125 g·mL^−1^	0.0625 g·mL^−1^
S1	454.67 ± 3.85	418.99 ± 9.55 ^###^	70.91 ± 5.14 ^&&&^	117.45 ± 1.30 ^※※※^	22.28 ± 0.93 ^$$$^
S2	278.83 ± 4.46 ***	21.58 ± 5.26 ^###^	64.59 ± 9.78 ^&&&^	45.47 ± 9.02 ^※※※^	37.18 ± 7.56 ^$$$^
S3	234.17 ± 5.81 ***	341.8 ± 8.01 ^###^	127.43 ± 9.78 ^&&&^	22.47 ± 2.62 ^※※※^	24.69 ± 1.44 ^$$$^
S4	233.08 ± 6.51 ***	626.31 ± 5.43	63.57 ± 6.30 ^&&&^	143.89 ± 6.49 ^※※※^	42.77 ± 1.96 ^$$$^
S5	329.53 ± 1.41 ***	440.58 ± 2.31 ^##^	316.60 ± 0.59	69.32 ± 8.33 ^※※※^	−106.26 ± 1.60 ^$$$^
S6	181.77 ± 8.71 ***	135.71 ± 8.83 ^###^	105.49 ± 5.91 ^&&&^	170.03 ± 3.29	105.84 ± 5.37
S7	12.09 ± 8.03 ***	173.41 ± 7.85 ^###^	69.94 ± 6.26 ^&&&^	119.68 ± 9.18 ^※※※^	120.83 ± 2.19
S8	−77.81 ± 3.75 ***	59.77 ± 5.90 ^##^	273.46 ± 9.67 ^&&&^	170.61 ± 8.61	60.21 ± 5.47 ^$$$^
S9	223.02 ± 9.39 ***	240.69 ± 2.98 ^###^	−334.22 ± 9.47 ^&&&^	−317.75 ± 7.6 ^※※※^	−233.97 ± 8.21 ^$$$^
S10	67.55 ± 3.21 ***	204.86 ± 9.18 ^###^	199.55 ± 0.12 ^&&&^	166.19 ± 10.51	80.28 ± 3.05 ^$$$^

Note: Compared with S1 (1 g·mL^−1^), *** *p* ≤ 0.001; Compared with S4 (0.5 g·mL^−1^), ^##^
*p* ≤ 0.01, ^###^
*p* ≤ 0.001; Compared with S5 (0.25g·mL^−1^), ^&&&^
*p*≤ 0.001; Compared with S8 (0.125 g·mL^−1^), ^※※※^
* p* ≤ 0.001; Compared with S7 (0.0625 g·mL^−1^), ^$$$^
*p* ≤ 0.001.

**Table 2 ijms-19-03439-t002:** Activation effects of the knocked-out components of water extract of *P. corylifolia* and negative samples on tyrosinase.

Peek No.	Activation Rate (%) on Tyrosinase Activity		
	Ethanol Extracts	Px+	Px−
P2	188.97 ± 0.13	52.19 ± 1.05	144.97 ± 1.32
P5	188.97 ± 0.13	−5.62 ± 3.98	294.96 ± 3.50
P6	188.97 ± 0.13	51.83 ± 2.50	222.51 ± 1.91
P13	188.97 ± 0.13	27.01 ± 0.87	23.42 ± 1.67
P14	188.97 ± 0.13	82.15 ± 2.68	3.73 ± 1.73
P18	188.97 ± 0.13	16.48 ± 2.56	132.10 ± 0.30
P20	188.97 ± 0.13	−161.24 ± 0.65	−176.37 ± 1.59
P23	188.97 ± 0.13	132.72 ± 2.38	227.12 ± 0.53
P24	188.97 ± 0.13	75.47 ± 3.60	200.79 ± 0.80
P25	188.97 ± 0.13	3.71 ± 0.40	116.78 ± 0.76
P27	188.97 ± 0.13	3.66 ± 0.41	−4.46 ± 0.23
P28	188.97 ± 0.13	18.03 ± 0.61	101.36 ± 0.53
P30	188.97 ± 0.13	−151.24 ± 0.68	−229.29 ± 0.60

**Table 3 ijms-19-03439-t003:** Information about the collected *P. corylifolia.*

Number	Collecting Land	Collecting Time
S1	Bozhou	December 2012
S2	Hebei	October 2010
S3	Yunnan	February 2015
S4	Yunnan	February 2016
S5	Yunnan	February 2017
S6	Henan	December 2013
S7	Imported	November 2013
S8	Imported	November 2014
S9	Imported	November 2015
S10	Imported	November 2016

**Table 4 ijms-19-03439-t004:** The time program of gradient elution.

Time/min	A/%	B/%
0–9 min	15%	85%
9–11 min	15–25%	85–75%
11–35 min	25–35%	75–65%
35–75 min	35–50%	65–50%
75–95 min	50–70%	50–30%
95–120 min	70–95%	30–5%

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
