# Peer review of "Screening the Marker Components in Psoralea corylifolia L. with the Aids of Spectrum-Effect Relationship and Component Knock-Out by UPLC-MS2"

_ijms, 2018, doi:10.3390/ijms19113439_

Round 1

Reviewer 1 Report

The paper presents screening the marker components in Psoralea corylifolia L. with the aids of spectrum-effect relationship and component knock-out by UPLCMS2.

The manuscript is suitable for International Journal of Molecular Sciences, yet below points need to be further explained or revised:

1) For what reason was maceration chosen as the extraction method? It would be useful to develop a modern technique, for example: microwave assisted extraction (MASE), accelerated solvent extraction (ASE).

2) How did the authors verify that the extraction was exhaustive?

3) Discussion should be enriched with comparison of the obtained results with other references.

4) The Authors should familiarize themselves with the proper format for References and make appropriate corrections. This section  is neglectfully prepared.

5) The English language, of the paper, needs to be improved. The work abounds in spelling errors and typos.

I can recommend this work for publication in International Journal of Molecular Sciences after minor revisions.

Author Response

Point 1: For what reason was maceration chosen as the extraction method? It would be useful to develop a modern technique, for example: microwave assisted extraction (MASE), accelerated solvent extraction (ASE). Response 1: Thank you for your comments. Maceration was chosen as the extraction method according to the technology of Psoralea corylifolia injection. In clinic, this injection is used to treat vitiligo. So, this method is used in our research in order to find the active compound for the Company to improve their technology. In the future, we may use the method of MASE, ASE to improve the technology of injection. Point 2: How did the authors verify that the extraction was exhaustive? Response 2: it's a good question. HPLC was used to detect the trace components to decide whether the extraction is exhaustive. Point 3: Discussion should be enriched with comparison of the obtained results with other references. Response 3: Thank you for your comments. We have made corresponding modifications and marked them in red in our manuscript. Point 4: The Authors should familiarize themselves with the proper format for References and make appropriate corrections. This section is neglectfully prepared. Response 4: Thank you for your suggestion. We have re-read the format for References in guidlines for authors and do the corresponding modification. Point 5: The English language, of the paper, needs to be improved. The work abounds in spelling errors and typos. Response 5: Thank you for your comments for our manuscript. We have checked the work abounds in spelling errors and typos and revised, pleased check it.

Reviewer 2 Report

Title of work is appropriate. This work has elements of scientific novelty. This is a good paper with acceptable explanation of the need of the method, performance and description of the methods and instruments, presentation of results, discussion of the results.

In my opinion, the above paper manuscript should be published in presented form in International Journal of Molecular Sciences

Author Response

Point 1: Title of work is appropriate. This work has elements of scientific novelty. This is a good paper with acceptable explanation of the need of the method, performance and description of the methods and instruments, presentation of results, discussion of the results. In my opinion, the above paper manuscript should be published in presented form in International Journal of Molecular Sciences. Response 1: Thank you for your admiration. We will take this article more seriously to ensure that this manuscript can be published more strictly in International Journal of Molecular Sciences